# The Importance of Physioxia in Mesenchymal Stem Cell Chondrogenesis and the Mechanisms Controlling Its Response

**DOI:** 10.3390/ijms20030484

**Published:** 2019-01-23

**Authors:** Girish Pattappa, Brian Johnstone, Johannes Zellner, Denitsa Docheva, Peter Angele

**Affiliations:** 1Laboratory of Experimental Trauma Surgery, Department of Trauma Surgery, University Hospital Regensburg, Franz Josef Strauss Allee 11, 93053 Regensburg, Germany; johannes.zellner@ukr.de (J.Z.); denitsa.docheva@ukr.de (D.D.); peter.angele@ukr.de (P.A.); 2Department of Orthopaedics and Rehabilitation, Oregon Health & Science University, 3181 SW Sam Jackson Park Rd, Portland, OR 97239, USA; johnstob@ohsu.edu; 3Sporthopaedicum Regensburg, Hildegard von Bingen Strasse 1, 93053 Regensburg, Germany

**Keywords:** mesenchymal stem cells, chondrogenesis, hypoxia, cartilage, hypertrophy, hypoxia inducible factors, early osteoarthritis

## Abstract

Articular cartilage covers the surface of synovial joints and enables joint movement. However, it is susceptible to progressive degeneration with age that can be accelerated by either previous joint injury or meniscectomy. This degenerative disease is known as osteoarthritis (OA) and it greatly affects the adult population. Cell-based tissue engineering provides a possible solution for treating OA at its earliest stages, particularly focal cartilage lesions. A candidate cell type for treating these focal defects are Mesenchymal Stem Cells (MSCs). However, present methods for differentiating these cells towards the chondrogenic lineage lead to hypertrophic chondrocytes and bone formation in vivo. Environmental stimuli that can stabilise the articular chondrocyte phenotype without compromising tissue formation have been extensively investigated. One factor that has generated intensive investigation in MSC chondrogenesis is low oxygen tension or physioxia (2–5% oxygen). In vivo articular cartilage resides at oxygen tensions between 1–4%, and in vitro results suggest that these conditions are beneficial for MSC expansion and chondrogenesis, particularly in suppressing the cartilage hypertrophy. This review will summarise the current literature regarding the effects of physioxia on MSC chondrogenesis with an emphasis on the pathways that control tissue formation and cartilage hypertrophy.

## 1. Introduction

Articular cartilage is a tissue lining the surface of synovial joints that provides friction-free movement and facilitates load-bearing. Chondrocytes are the specialised cells within the tissue that create and maintain cartilage matrix that is primarily composed of aggrecan (a large proteoglycan) and collagen II. Chondrocyte orientation and matrix distribution within the tissue generates an anisotropic structure from the cartilage surface to the deep zone and outwards from the chondrocyte. These cartilage zones are described as the superficial, middle/transitional, deep and calcified zones; whilst the pericellular, territorial and interterritorial regions surround the chondrocyte. The differences in matrix distribution within these zones and regions contribute to cartilage biomechanics, enabling it to withstand high dynamic and compressive loads during joint loading [1,2].

However, articular cartilage is susceptible to progressive degeneration leading to changes in its structure and function. This degenerative disease is described as osteoarthritis (OA). Early observable events in OA include collagen fibrillation of the superficial layer that leads to loss of proteoglycans and other matrix molecules. Chondrocytes begin to proliferate and form clusters with increased matrix synthesis in response, whilst inflammatory cytokines stimulate the expression of matrix metalloproteinases (MMPs) and aggrecanases (ADAMTS) that facilitate cartilage degradation [3,4]. Eventually chondrocytes reduce their proliferative and anabolic response and the degeneration continues, whilst pain and changes in joint function become apparent and are clinical indicators for OA.

It has been noted that there is an increased risk of OA within the knee joint due to previous joint injury (e.g., anterior cruciate ligament (ACL) injury), excessive repetitive loading, joint dysplasia and meniscectomy. According to the German Cartilage Registry (Deutsche Gesellschaft für Orthopäedie und Unfallchirurgie (DGOU)) between October 2013–June 2014, 60% of treated cartilage defects were degenerative, whilst a recent multi-centre study showed that of 400 patients, approximately 40% had chondral injuries resulting from degenerative conditions [5,6]. In classifying the forms of OA that can be treated, recent studies have described the term “early OA” [7,8,9]. This latter condition may be amenable to regenerative medicine or tissue engineering therapies. An example of cartilage tissue engineering is autologous chondrocyte implantation (ACI) that is currently being used to treat focal cartilage defects. However, a high failure and re-operation rate has been observed when treating focal degenerative lesions compared with post-trauma defects. One potential reason for their poor outcome is the surrounding inflammatory environment that impairs cell-based solutions. In particular, IL-1β has been shown to be a negative predictor for ACI treatment post-surgery [5,10]. Thus, a primary goal for clinicians and scientists is to develop regenerative options that can be used to treat both focal and diffuse early OA in this challenging environment.

Autologous articular chondrocytes are an established cell-based tissue engineering strategy for treatment of large “focal traumatic” or “focal early OA” chondral or osteochondral defects of the knee joint [6,11,12]. Mesenchymal stem cells (MSCs) can be isolated from a variety of sources including bone marrow, adipose (liposuction or intrapatellar fat pad) or synovium with minimal donor site morbidity [13,14,15,16] and have been shown to have chondrogenic potential, initially in vitro with the creation of pellets or micromasses and the addition of the stimulatory growth factor, transforming growth factor-beta (TGF-β) [17,18]. Scaffolds/biomaterials have been developed in which MSCs are seeded or encapsulated, and then chondrogenically differentiated to create clinically relevant chondrogenic implants that may be used to fill patient defects. However, in both pellets and scaffolds, markers of chondrocyte hypertrophy (collagen X, MMP13) have been detected and upon implantation in vivo, ectopic bone formation can occur [19]. Strategies to prevent hypertrophy and produce a stable articular chondrocyte phenotype with its defined extracellular matrix are the principle goals of in vitro cartilage tissue engineering.

Scientists have attempted to attain stable cartilage formation with environmental stimuli relevant to the in vivo situation, e.g., biomechanical stimulation, lower oxygen tension and/or addition of appropriate growth factors. The rationale for using low oxygen tension for cartilage tissue engineering is the oxygen level within articular cartilage that ranges from 2–5% oxygen (Figure 1) [20,21,22,23,24]. The use of growth factors (e.g., TGF-β), low oxygen tension and other stimuli for MSC chondrogenesis can also be used to cause the redifferentiation of articular chondrocytes in either pellets or hydrogels. Adding TGF-β under physioxia upregulates chondrogenic gene (*sex-determining region Y–box 9* (*SOX9)*, *collagen II alpha I* (*COL2A1)* and *aggrecan* (*ACAN*)) expression and subsequent glycosaminoglycan (GAG) and collagen II accumulation, whilst downregulating the expression of catabolic genes (e.g., *MMP9*, *MMP13)*, *ADAMTS-4*, *ADAMTS-5*) compared with cells in hyperoxic (atmospheric) conditions [25,26,27,28,29,30,31,32]. Similarly, OA chondrocytes cultured in the same low oxygen conditions also demonstrated an increase in GAG deposition and a reduction in the expression of MMPs, although they had higher *collagen X alpha I* (*COL10A1*) and *MMP13* expression and lower GAG deposition compared with healthy chondrocytes [30,31]. Additionally, low oxygen culture can also counter the inhibitory effects caused by the presence of inflammatory cytokines [29]. Thus, lowered oxygen tension could potentially help to induce a stable chondrogenic phenotype in MSC chondrogenesis, and many studies have been conducted to understand its effects on this process. The present review will summarise the literature and evaluate the effects of low oxygen tension or physioxia on MSC chondrogenesis, evaluating the outcomes of the various studies and the pathways that have been identified to be part of the cellular response to it. A Pubmed search was performed; the date of the last search was 31 October 2018. The keywords used for the gathering of relevant publications used the terms, “mesenchymal stem cells” and “hypoxia” or “chondrogenesis” or “chondrocytes”. Publications since 2001 were evaluated for the purposes of this review.

## 2. Physioxia and Cartilage

In vivo physiological oxygen tension (physioxia) within human articular cartilage ranges from 2–5%, whilst bone marrow physioxia has been measured at 7% oxygen (Figure 1) [20,21,23,24]. The atmospheric oxygen level (20% oxygen) at which most typical tissue culture incubators operate is actually non-physiological and represents hyperoxia [20,33].

In all of the studies summarised for this review, incubator oxygen tension (20%; hyperoxia) is utilised as the ‘control’ for experiments done at physioxia even though the latter is closer to conditions found in vivo. An alternative method to mimic some of the effects of physioxia is through chemical induction, specifically on the pathway controlling the stability of *hypoxia-inducible factor-1 alpha* (*HIF-1α*) that has been found to be essential for chondrocyte survival and cartilage homeostasis (Figure 2) [34,35]. In chemically-induced physioxia, cobalt chloride is the most extensively studied compound. Under hyperoxia, *HIF-1α* is hydroxylated by prolyl hydroxylases (PHDs) and factor inhibiting HIFs (FIH), resulting in its proteosomal degradation (Figure 2a). The process is catalysed by molecular oxygen, iron ions, ascorbic acid and 2-oxoglutarate [36,37,38]. Cobalt chloride competes with iron ions for the active site of PHD and prevents *HIF-1α* hydroxylation, thereby stimulating a physioxic response in hyperoxic oxygen conditions. Other hydroxylase inhibitors have also been used to create a physioxia mimicking response, specifically dimethyloxalylglycine (a competitive inhibitor of PHDs and FIHs) and desferrioxamine (sequesters iron ions). Although this review primarily focusses on experiments that use altered incubator oxygen levels, studies utilising chemically-induced physioxic responses are discussed where appropriate [36,39,40].

## 3. MSC Isolation and Expansion under Physioxia

Investigators examining MSC proliferation under physioxia have demonstrated that the initial plating of human bone marrow produced greater colony-forming units–fibroblasts (CFU-Fs) upon culture in a low oxygen environment [41,42,43,44]. Furthermore, MSCs were able to proliferate at a shorter cell doubling time and have consequent increased cell numbers in physioxia [43,44,45,46]. This pattern has also been replicated in the majority of studies for chondrogenic cells from different tissue sources (adipose and synovium) and other species (porcine, ovine and murine) [43,45,47,48,49,50,51,52,53].

Long-term culture under lowered oxygen conditions caused MSCs to reach greater population doublings with reduced cellular senescence [46,52,54,55]. The latter phenomena has been postulated to be due to the underlying cellular ATP metabolism. Under hyperoxia, MSCs have an increased oxygen consumption, thereby having a greater ATP production via oxidative phosphorylation. In comparison, physioxic MSC cultures had greater lactate production and reduced oxygen consumption compared with hyperoxic MSC cultures, resulting in greater energy production via glycolysis [54,56,57]. The increased utilisation of oxidative phosphorylation under hyperoxia generated reactive oxygen species that can enhance cellular senescence [58]. Furthermore, telomeres that control cellular aging were found to be longer for physioxia-cultured MSCs [46]. These investigations demonstrate a clear advantage of physioxia for MSC expansion.

Studies have also demonstrated that stem cell markers that are typically associated with either embryonic or induced pluripotent stem cells (e.g., *octamer-binding transcription 4* (*OCT-4)*, *stage-specific embryonic antigen* (*SSEA-1*) are expressed on MSCs expanded in physioxia [59,60]. Consistent with this, physioxia-expanded MSCs can undergo osteogenesis and adipogenesis at later population doublings compared with those expanded in hyperoxia. However, there are also studies showing that osteogenesis is inhibited under physioxia and with physioxia expanded MSCs differentiated under hyperoxia, indicating that differences in oxygen tension can affect differentiation towards different musculoskeletal lineages [48,54,59]. The present review focusses on how physioxia affects MSC chondrogenesis in terms of the amount of matrix formation that occurs and the types of matrix molecules that are expressed. The influence on the latter is of particular interest, as hyperoxic in-vitro MSC chondrogenesis results in the expression of hypertrophic markers (e.g., collagen X, MMP13, alkaline phosphatase) that upon implantation in a nude mouse model result in ectopic bone formation in vivo [19].

## 4. Physioxia and MSC Chondrogenesis

### 4.1. Chondrogenic Matrix Formation

A summary of the outcomes for the effects of physioxia on MSC chondrogenic matrix formation are described in Table 1 and Table 2. Mouse stromal cells in a pellet culture model demonstrated that physioxia (1% oxygen) upregulated *SOX9* gene expression and increased GAG deposition in the matrix [61]. Later studies were in agreement with the described investigation, whilst also observing an increase in pellet wet weight and an upregulation in *COL2A1* and *ACAN* expression [46,62]. However, there are also studies that found a downregulation in matrix gene expression (*SOX9*, *COL2A1*, *ACAN*) and no effect on matrix formation [44,51,53,54,63,64,65,66,67,68,69]. One study using adipose-derived murine MSCs cultured under physioxia demonstrated a reduction in both GAG and collagen II matrix content compared to hyperoxia, in spite of larger pellet diameter [66]. It was assumed in these cases that oxygen gradients had formed under hyperoxia, whereby a physioxic region develops within the central regions of pellets or scaffolds that correspond to greater GAG and collagen II matrix deposition [63,70].

Donor variability has been shown to influence chondrogenic potential, especially in response to low oxygen culture. Anderson et al. [33] investigated the effect of physioxia on MSC preparations with different levels of chondrogenic potential. Their intrinsic chondrogenic capacity was based on the amount of GAG produced relative to that formed by articular chondrocytes formed under normal atmospheric conditions. It was demonstrated that in preparation with low GAG, there was enhanced chondrogenesis under physioxia, whilst high GAG producing MSC preparations did not get a significant enhancement from lower oxygen. Thus, the differences described in studies displaying no influence of physioxia on MSC chondrogenesis could be related to the donor variability in chondrogenic potential.

With this as a caveat, the majority of studies of MSC chondrogenesis either in pellet or scaffold cultures indicates an increase in chondrogenic genes (*COL2A1* and *ACAN*) and matrix formation under physioxia [33,41,42,43,47,49,50,52,55,62,71,72,76,77,78,79,82,86,87,88,90,92,93,95,97]. Furthermore, it has also been shown that physioxia is a more potent promoter of chondrogenesis than dynamic (compressive) loading with no synergistic effect upon combining the parameters [78,98].

### 4.2. MSC Hypertrophy

The effect of physioxia on chondrogenic hypertrophy has been a focus of many studies. A downregulation in hypertrophic (*COL10A1*) gene expression has been measured in the majority of studies [33,42,43,47,51,63,65,69,70,80,82,83,84,87,92,93,95], although there is some recorded upregulation [62,67,72] in this hypertophic marker. It has also been shown that the timing of physioxic culture influences this process. Culture under physioxia after an initial 2 weeks of hyperoxic culture significantly reduced the expression of hypertrophic genes and increased that of hypertrophy antagonists, Gremlin-1 (GREM1), Frizzled-related protein (FRZB) and Dickkopf WNT (DKK1). These antagonists were reduced and hypertrophic genes were upregulated upon hyperoxic culture [82].

As previously discussed, Anderson et al. [33] demonstrated the importance of low GAG and high GAG MSC preparations on chondrogenic matrix production. Subsequent analysis of hypertrophy genes indicated that despite a downregulation of *COL10A1* in both MSC preparations, only high GAG MSC donors expressed a significant difference relative to hyperoxic conditions. However, collagen X staining was observed in physioxia-treated pellets independent of whether they were high or low GAG MSCs. This also demonstrates differences in gene expression and protein data with respect to hypertrophy, and this has been shown in a few studies [33,42,82,99]. Thus, physioxia suppresses collagen X expression but does not completely inhibit its production.

A few studies have shown that physioxia also downregulates collagenases (*MMP13*) and aggrecanases (*ADAMTS-4* and *ADAMTS-5*) involved in cartilage hypertrophy. Examination of osteogenic gene expression (*runt-related transcription factor-2* (*RUNX2*), *osteopontin* and *osteonectin*) and calcium deposition within the cartilaginous matrix was also found to be inhibited under physioxia [45,83,91]. In this instance, inhibited osteogenesis may be related to an underlying metabolism that favours a glycolytic metabolism, which is a critical component for sustaining an articular chondrocyte phenotype and thus, may also operate during MSC chondrogenic differentiation [57,59,71]. A microarray analysis showed an upregulation in glycolysis pathway-associated enzymes under physioxia, indicating its importance in developing a stable chondrocyte phenotype [82].

### 4.3. MSC Preconditioning and In Vivo Implantation

MSC that have been pre-expanded or preconditioned under physioxia prior to subsequent chondrogenesis under the same conditions demonstrated an enhancement in matrix deposition and a reduction in collagen X and other hypertrophy markers compared with non-preconditioned chondrogenic MSCs differentiated under physioxia (summarised in Table 2) [41,42,43,45,46,48,49,52,62,89,90,91,93,94,95,97]. Martin-Rendon et al. (2007) showed that preconditioned MSCs had higher *SOX9* gene expression and subsequent pellet wet weight [89]. Physioxic preconditioned MSCs have demonstrated an increase in gene expression and cartilage matrix formation, even when differentiated under hyperoxia, when compared with non-preconditioned MSCs [42,43]. Examination of physioxia isolated MSCs has been shown to have a significantly reduced expression in CD90 or Thy-1 expression under these conditions [42]. Articular chondrocytes have been shown to not express CD90 in vivo but it is expressed during in vitro culture, concomitant with a loss of chondrogenic potential [100]. Potentially, a pre-selection process occurs under physioxia, whereby a chondroprogenitor population is isolated. It might be possible to use CD90 or other cell surface markers to evaluate whether physioxia can select for MSCs that have a high chondrogenic potential.

Subsequent implantation of physioxia-conditioned chondrogenic MSCs in vivo has been shown to increase cartilage matrix formation (collagen II and GAG) compared with implanted hyperoxic-conditioned MSCs. Furthermore, physioxia-preconditioned MSCs demonstrated reduced bone-like formation in both nude mouse and rabbit models, unlike previous investigations using hyperoxic MSCs in nude mouse models [81,82,92]. However, in larger animal models, despite an increase in in vitro chondrogenic gene expression and matrix formation, subsequent in vivo implantation provided no significant advantage relative to hyperoxic MSCs [51]. The sustained low oxygen environment in vivo could eliminate the difference between physioxic and hyperoxic preconditioning that is seen in short-term in vitro cultures. Further studies are required to understand whether physioxic preconditioned MSCs or chondrogenic implants are beneficial for in vivo chondrogenesis, specifically in larger animal models.

### 4.4. Physioxia Prevents Cytokine Inhibited Chondrogenesis

To mimic the OA environment and evaluate the performance of MSC chondrogenic implants for the treatment of focal defects, inflammatory cytokines have been used. IL-1β and TNF-α have previously been demonstrated to inhibit MSC chondrogenesis under hyperoxia [74,85]. An investigation showed that the reduction in matrix formation in the presence of IL-1β was suppressed by physioxic culture [74]. In particular, the matrix degradative enzymes (MMP1, MMP3 and MMP13) induced by IL-1β presence were reduced under physioxia. A recently published study reported similar results, but it was noted that physioxia does not completely restore chondrogenesis to control levels in the presence of IL-1β, and only low GAG MSC donors or physioxia responders displayed a significant increase in chondrogenic gene expression and cartilage matrix formation [99].

A potential reason for the restoration of the chondrogenic phenotype in the presence of IL-1β may be related to the upregulation in TGF-β receptors I and II under physioxia chondrogenesis. In this instance, chondrocytes in the presence of IL-1β have been shown to impair matrix production through downregulating TGF-β receptor II expression, whilst upregulating the NF-κB pathway [101,102,103]. This hypothesis was supported by a recent investigation, whereby there was an upregulation in TGF-β receptor I and II expression in physioxia MSC chondrogenesis in the presence of IL-1β compared to a downregulation in these receptors in the corresponding hyperoxic condition [99].

As well as the documented increase in inflammatory cytokines, the process of OA has also been found to increase in oxygen tension within the joint [22]. Reoxygenation of physioxia MSCs mimics the changes in oxygen tension during OA. In vitro experiments demonstrated that reoxygenating MSC chondrogenic pellets in the presence of TNF-α showed an increase in aggrecanolysis with subsequent upregulation in *ADAMTS4/5* and *MMP* expression [85]. Thus, physioxia helps to suppress inflammatory cytokine-inhibited MSC chondrogenesis via suppression of matrix degradative enzymes and restoration of TGF-β receptors [99].

## 5. Physioxia Mechanisms in MSC Chondrogenesis

### 5.1. Hypoxia Inducible Factors (HIF): HIF-1α

Investigations of the mechanism controlling the chondrogenic response under physioxia have noted the importance of the *hypoxia-inducible factor* (*HIF*) genes. The HIF genes belong to the basic helix-loop-helix (bHLH)-Per-Ant-Sim (PAS) family of transcription factors [104,105,106]. There are three HIF genes, *HIF-1*, *HIF-2* and *HIF-3*, which consist of two sub-units, an unstable oxygen-sensitive alpha sub-unit and a stable oxygen-insensitive beta sub-unit. The most studied is *HIF-1α*, which has been described to be essential in cartilage development [34,35]. Under hyperoxic conditions, *HIF-1α* is hydroxylated by PHDs and this exposes 4-hydroxyproline residues on the molecules. These exposed residues are recognised by the von-Hippel-Lindau (VHL) E3 ubiquitin ligase complex, resulting in its proteasomal degradation. Alongside this process, FIH hydroxylates the asparagine residue on HIF-1α, thereby preventing its binding to the co-factor p300 (Figure 2a). In contrast, physioxia inhibits PHD and FIH hydroxylation, enabling *HIF-1α* to translocate the nucleus and dimerize with *HIF-1β* (Figure 2b).

*HIF-1α* activation upregulates the expression of physioxia-sensitive genes that include the cartilage transcription factors, *SOX5*, *SOX6* and *SOX9*, plus matrix genes, *COL2A1* and *ACAN* [72,73,77,92]. It has been shown that *HIF-1α* interacts with *SOX9* to upregulate *COL2A1* and *ACAN*, whilst downregulating *COL10A1* expression under physioxia [92]. However, using either cadmium chloride (chemical physioxia inhibitor) or *HIF-1α* dominant negative plasmid on physioxia-cultured chondrogenic MSCs reduced *SOX9*, *COL2A1* and *ACAN* expression, whilst *COL10A1* expression was increased. The importance of *HIF-1α* was further supported in a recent study, whereby chondrogenically differentiating *HIF-1α* deleted murine adipose MSCs under physioxia and developed micromasses with reduced GAG and collagen II content compared to controls [73]. Furthermore, an analysis of different *HIF-1α* hydroxylase inhibitors that chemically-induced physioxia effects under hyperoxia via *HIF-1α* stabilisation was also performed [36]. Dimethyloxalylglycine (DMOG), a competitive inhibitor of PHD and FIH, was found to induce the most stable articular chondrocyte phenotype in MSC chondrogenesis compared with other forms of chemically-induced physioxia. Application of the HIF-1α/-β complex inhibitor, acriflavine, on DMOG induced MSC chondrogenesis and downregulated *SOX9* and *COL2A1* gene expression, whilst upregulating *COL10A1* and increasing hypertrophic chondrogenesis. This indicates that stable *HIF-1α* is an important component involved in the physioxic chondrogenic response (Figure 3a).

Additionally, stable *HIF-1α* expression has been demonstrated to induce genes associated with the glycolysis pathway both in articular chondrocytes and chondrogenic MSCs [57,71,82]. Articular chondrocytes have highly glycolytic metabolism due to their in vivo environment and they undergo cellular senescence upon hyperoxic culture due to the production of reactive oxygen species [54,55,58]. Chondrogenic MSCs have also shown the same cellular metabolism during differentiation, and physioxia helps sustain this glycolytic metabolism. Thus, activation and stabilisation of the *HIF-1α/HIF-1β* complex could be a factor in achieving a stable articular chondrocyte phenotype.

### 5.2. HIF-2α

*HIF-2α* or *EPAS-1* has been investigated in chondrocytes and MSCs with regards to its expression both in physioxia and OA cartilage [107,108,109,110]. Coimbra et al. [109] demonstrated that *HIF-2α* was expressed on an mRNA level in both normal and osteoarthritic chondrocytes under hyperoxic conditions. This result has been replicated in a recent investigation, whereby *HIF-1α* and *HIF-2α* were expressed under hyperoxic and physioxic conditions in both healthy and OA chondrocytes [31].

*HIF-2α* expression in hypertrophic and osteoarthritic chondrocytes has been a major focus area with contrary results regarding its function in these particular processes. In embryonic mouse tibial cartilage, *HIF-2α* expression was located in the hypertrophic zone alongside chondrocytes expressing hypertrophic markers, *COL10A1* and *MMP13* [110]. *HIF-2α* overexpression in a mouse chondrocyte cell line (ATDC5) resulted in an upregulation of hypertrophic, OA-associated (e.g., *COL10A1*, *MMP13* and *vascular endothelial growth factor* (*VEGF*)) and osteogenic (e.g., *RUNX2*, *osteopontin*) genes. Furthermore, an analysis of healthy and OA cartilage from human donors indicated greater *HIF-2α* expression with increasing OA grade. The results were also supported by an independent study (Figure 3b) [107].

These studies also included mouse models to understand whether these in vitro effects were observed in vivo [107,110]. In one study, transgenic *HIF-2α* overexpressing mice using promoter and enhancer regions of mouse *COL2A1* gene were developed to understand the in vivo response [107]. With age, there was increased cartilage degradation with the concomitant expression of MMPs, aggrecanses and downstream catabolic targets, nitric oxide synthase (NOS2) and prosteoglandin-endoperoxide synthase-2 (PTGS2). Saito et al. (2010) [110] generated heterozygous *HIF-2α* knockout (*HIF-2α^+/−^*) mice that had a larger hypertrophic zone with a reduced bone area compared to wild type mice, indicating that *HIF-2α* insufficiency prevents chondrocyte hypertrophy and subsequent matrix degradation and vascularization. Furthermore, use of either collagenase injection or a displaced medial meniscus (DMM) model to induce osteoarthritis in these heterozygous mice demonstrated no cartilage degeneration and a suppression of MMP and aggrecanase expression and downstream catabolic targets, NOS2 and PTGS2 [107]. Thus, these models indicate that *HIF-2α* expression results in cartilage destruction, whilst its inhibition prevents OA and endochondral bone development. Studies investigating the effects of inflammatory cytokines (IL-1β and IL-6) on mouse articular chondrocytes showed that an increase in *HIF-2α* expression induced cartilage destruction via the downstream activation of the NF-κB pathway, a well-described OA mechanism [107,111].

However, these early publications have been questioned by recent investigations that replicated the conditional *HIF-2α* knockout mouse model and demonstrated that it only induced a delay in endochondral ossification with no change in *COL10A1* expression and no correlation in human subjects between *HIF-2α* expression and idiopathic OA [112,113,114]. Furthermore, in-vitro physioxic culture of human articular chondrocytes or chondrogenic MSCs has been shown to upregulate *HIF-2α* and subsequently promotes cartilage matrix gene expression and formation [26,42,72]. A commentary suggested that differences in mouse and human cartilage, particularly with respect to cartilage loading and thickness, mean that *HIF-2α* in humans and larger animals has evolved into an anabolic gene rather than a catabolic gene described in mouse studies [114]. Additionally, the in vitro experiments examining *HIF-2α* overexpression were conducted under hyperoxia; thus, it is not known how this would be influenced under physioxia. Therefore, the function of *HIF-2α* remains elusive and further studies are required in both chondrocytes and chondrogenic MSCs under physioxia to understand its contribution to chondrogenesis and cartilage homeostasis.

### 5.3. HIF-3α

The third isoform, *HIF-3α*, has only recently been investigated in chondrogenesis [115,116,117]. Articular chondrocytes upon redifferentiation expressed *HIF-3α* micromass pellet cultures with greater expression under physioxia, although it had a lower gene expression level compared to *HIF-1α* and *HIF-2α* under the same conditions [115]. A reason for the differences in transcript levels may be related to only a small number of chondrocytes expressing the gene. However, the fact that it was highly expressed in physioxic cultures and corresponded to the upregulation in cartilage matrix genes during redifferentiation suggested that it was contributory to the chondrocyte phenotype. Furthermore, there is evidence that upregulated *HIF-3α* expression under physioxia inhibits *COL10A1* and *MMP13* expression, whilst knockdown of this gene upregulates these OA markers under the same conditions in a Ewing sarcoma cell line [116].

A recent study investigated *HIF-3α* expression in articular chondrocytes and chondrogenic MSCs under physioxia and hyperoxia to understand whether its expression was associated with preventing hypertrophy [117]. Chondrogenic MSCs under physioxia and hyperoxia showed a low HIF-3α expression, whilst there was higher expression for *COL10A1* and *MMP13*. In contrast, redifferentiated articular chondrocytes had a higher *HIF-3α* expression and downregulation in *COL10A1* and *MMP13*, independent of the oxygen condition. Furthermore, OA chondrocytes were found to have a downregulation in *HIF-3α* expression compared with healthy articular chondrocytes. Analysis of human embryonic growth plate cartilage demonstrated that *HIF-3α* expression was greatest in the precursor chondrocyte region, whilst hypertrophic chondrocytes exhibited minimal *HIF-3α* expression but had maximal expression for *COL10A1* and *MMP13* compared with other regions within the growth plate [117]. These findings indicate that *HIF-3α* is a regulator of chondrocyte hypertrophy, whereby its expression prevents hypertrophic gene expression and is diminished in cells with an upregulation in hypertrophic genes such as OA chondrocytes and chondrogenically differentiated MSCs (Figure 3c).

Investigations into *HIF-3α* expression, specifically in the previously described high and low GAG MSC donors could lead to an understanding of how *HIF-3α* can control MSC chondrogenic hypertrophy especially under physioxia. Furthermore, there is evidence that *HIF-3α* expression inhibits downstream targets of *HIF-1α* and *HIF-2α*. Based on investigations on articular chondrocytes and the described literature, it has been postulated that *HIF-3α* stabilisation inhibits the catabolic effects exerted by *HIF-2α* under physioxia [117]. The inter-relationship between these HIF genes in physioxia MSC chondrogenesis requires further investigation.

### 5.4. PI3K/Akt/FOXO Pathway

A process that has been gradually investigated in OA research is the process of apoptosis, as this contributes to the initiation of the disease via chondrocyte cell death. Furthermore, the classical process of endochondral ossification has been thought to involve the apoptosis of hypertrophic chondrocytes that initially express *COL10A1* and *RUNX2*, although there is now an evolving paradigm that is contrary to this process and involves the transdifferentation of chondrocytes into osteoblasts [118,119,120]. Thus, anti-apoptotic pathways are a potential mechanism for preventing chondrogenic MSC hypertrophy.

One relevant mechanistic pathway that has been investigated in MSC chondrogenesis is the PI3K/Akt pathway [93,121,122,123,124]. Transgenic mice containing an Akt fusion protein that enables constitutive Akt activation in articular cartilage resulted in highest Akt phosphorylation in resting and proliferative chondrocytes but reduced expression in hypertrophic chondrocytes [121]. An embryonic limb explant model using limbs from wild type and transgenic Akt mice was used to understand the significance of the PI3K/Akt pathway with respect to cartilage development. Following 5 days of culture, Akt activation increased the proliferative and reduced the hypertrophic zone within the limb. Use of a PI3K inhibitor (LY294002) on wild-type limbs only, reversed this effect and increased the hypertrophic zone within cartilage. These explant results were replicated in vitro, using human synovial stromal cells (hSSCs) in a pellet culture model. Lentivirally transduced hSSCs that expressed *Akt* gene via application of 4-hydroxytamoxifen, had increased pellet size and GAG deposition that corresponded with elevated *SOX9*, *COL2A1* and *ACAN* expression. Gene expression of *COL10A1* and *RUNX2* was downregulated under these conditions. However, inhibition of PI3K using LY294002 in hSSCs undergoing chondrogenesis demonstrated an inverse effect, whereby there was decreased pellet size and GAG deposition, whilst there was an upregulation in hypertrophy genes. Thus, this pathway contributes to the control of hypertrophy during chondrogenesis and has been studied further [122].

The PI3K/Akt pathway was specifically investigated in MSC chondrogeneis to understand whether it is involved in the suppression of hypertrophy under physioxia [93]. A downregulation in *COL10A1* and *RUNX2* gene expression and reduced collagen X staining was observed for physioxia MSC chondrogenic pellets. They also describe an increase in the percentage of apoptotic cells (active caspase-3 and -8) within chondrogenic pellets under hyperoxia compared with physioxia. Western blot analysis confirmed that Akt was phosphorylated in chondrogenic pellets under physioxia. Use of the PI3K inhibitor (wortmannin) in physioxia chondrogenic pellets showed that blocking downstream Akt phosphorylation resulted in a downregulation in chondrogenic gene expression (*SOX9*, *COL2A1*, *ACAN*) and an upregulation in *RUNX2* and *COL10A1* gene expression. These data were supported by immunohistochemical staining whereby there was increased collagen X and reduced GAG and collagen II staining in wortmannin treated pellets. Furthermore, there was a greater percentage of apoptotic cells (active caspase-3 and caspase-8) within physioxia chondrogenic pellets in the presence of the inhibitor.

One of the downstream targets of the PI3K/Akt pathway is the *Forkhead-box class O* (*FOXO*) transcription factors that consist of four members, *FOXO1*, *FOXO3*, *FOXO4* and *FOXO6* [125,126]. *FOXO1* and *FOXO3* were expressed within normal articular cartilage, localised within the superficial zone and middle zone of the tissue. However, phosphorylation of *FOXO1* and *FOXO3* provokes nuclear export into the cytoplasm and subsequent degradation [123]. Previous studies have shown that *FOXO1* and *FOXO3* phosphorylation within cartilage leads to cell death and osteoarthritis due to the loss of autophagy genes that are known to protect chondrocytes from stress and are under the control of *FOXO3* [127,128]. This was demonstrated in human cartilage, as OA cartilage had reduced expression for both *FOXO1* and *FOXO3* in the superficial zone compared to healthy cartilage, but had greater expression within the middle zone due chondrocyte clustering and phosphorylation which changed from the nucleus to the cytoplasm [123]. The importance of *FOXO* genes in articular cartilage homeostasis was further demonstrated, as *FOXO1* and *FOXO3* knockout mice developed more severe age-related OA compared with control mice, whilst the expression of autophagy genes was downregulated in triple knockout (*FOXO1*, *-3* and *-4*) mice [124]. Furthermore, adenoviral expression of *FOXO1* in human OA chondrocytes upregulated autophagy genes and countered the detrimental effects of IL-1β. Thus, the FOXO genes are important contributors to cartilage homeostasis.

In physioxic MSC chondrogenesis, *FOXO* transcription factors with respect to cartilage hypertrophy have also been investigated [93]. *FOXO1/FOXO3* was found to be expressed in physioxic chondrogenesis. However, inhibition of these transcription factors either via leptomycin B or 4-Hydroxynoneal replicated the response for physioxia chondrogenesis in the presence of wortmannin, whereby there was downregulation in chondrogenic gene expression, an upregulation in *RUNX2* and *COL10A1* gene expression with an increase in collagen X staining within these pellets. Furthermore, markers of apoptosis, active caspase-3 and caspase-8, were found to increase in physioxia chondrogenic pellets in the presence of FOXO inhibitors.

Thus, the PI3K/Akt/FOXO pathway (Figure 3d) may be a critical pathway in chondrogenesis with respect to controlling the suppression of hypertrophy and cartilage homeostasis. This pathway also involves *HIF-1α* and therefore the interactions between these transcription factors would require further investigation. Furthermore, how *FOXO* expression upregulates autophagy genes that help to protect against cartilage damage has not been determined, particularly with respect to MSC chondrogenesis in the presence of inflammatory cytokines (e.g., IL-1β) and the mechanism countering its inhibition under physioxia.

## 6. Summary

In general, physioxia has an anabolic effect on MSC chondrogenesis whereby there is an upregulation in chondrogenesis genes (*SOX9*, *COL2A1*, *ACAN*) and a concomitant increased matrix deposition both within pellets and scaffolds when compared with hyperoxic conditions. In contrast to hyperoxic culture, physioxic chondrogenesis significantly downregulates the expression of hypertrophic genes, *COL10A1* and *MMP13*, although subsequent staining suggests that physioxia does not completely suppress collagen X production, particularly for donors whose cells produce a high amount of GAG (Table 1 and Table 2). Implantation of physioxia preconditioned chondrogenic MSCs in nude mice and rabbit models replicated in vitro results in stable cartilage formation with reduced bone formation compared with hyperoxic MSC chondrogenic implants [81,82,92]. However, a similar study using a sheep model resulted in no differences in cartilaginous repair between physioxic and hyperoxic preconditioned chondrogenic implants [51]. More careful selection of MSC preparations may be required, as in spite of the downregulation of *COL10A1* and *MMP13* in high and low GAG donors, collagen X staining was present in both donor types under physioxia, indicating only the suppression of hypertrophy [33]. To completely inhibit the hypertrophy in MSC chondrogenesis, additional environmental stimuli are required to stabilise the chondrogenic phenotype. Mechanical stimuli has been known to enhance cartilage matrix production in MSC chondrogenesis [129,130,131]. One form of stimulation is hydrostatic pressure that involves fluid pressurisation without cellular deformation [132,133]. Theoretical models have described that this form of loading can help maintain the chondrocyte phenotype and prevent hypertrophy [134,135]. However, there are contrary results in the literature; in some studies, hydrostatic pressure prevented MSC hypertrophy, but there are also publications indicating an upregulation in hypertrophic genes [136,137,138,139,140,141,142]. It would be of interest to investigate the combination of physioxia and hydrostatic pressure during MSC chondrogenesis.

MSCs and chondrocyte co-culture models have been tested to examine the effects on the two cell types [67,143,144,145]. It was hypothesized that the cross-talk between cell types could help to improve the cartilage tissues for in vivo implantation. It has been shown that direct MSC-chondrocyte co-cultures prevent cartilage hypertrophy [143]. It has been demonstrated that co-culturing MSCs and chondrocytes under physioxia helped to increase both cartilage formation and reduce hypertrophy compared to MSC monocultures under the same conditions [67]. Further investigations into MSC-chondrocyte co-culture systems under physioxia, possibly with loading, are warranted.

Physioxia helps counter the detrimental effects of inflammatory cytokines during in-vitro chondrogenesis [74]. However, the underlying mechanisms controlling the response have not been established. The HIFs have been the primary focus of the majority of studies on this phenomenon. It appears that *HIF-1α* promotes the upregulation of chondrogenic matrix gene expression and *HIF-3α* may help to stabilise the chondrogenic phenotype [34,35,105,117]. In contrast, some studies indicate that *HIF-2α* upregulates both hypertrophic genes and matrix degradative enzymes [107,108,110,111]. However, there is also evidence that *HIF-2α* is also involved in the chondrogenic anabolic response in both MSCs and chondrocytes under physioxia [26,42,72]. Further investigations are required to understand the function of *HIF-2α* in the MSC chondrogenic response in physioxia, particularly in the regulation of inflammatory cytokines, as a means to using the HIF-related pathways to modulate the chondrocyte phenotype.

The PI3K/Akt/FOXO pathway (Figure 3d) is involved in preventing age-related OA; ectopic *FOXO* expression in OA chondrocytes in the presence of IL-1β upregulated genes associated with autophagy, thus countering the inhibitory effects of IL-1β [124]. This pathway has also been shown to be involved in reducing apoptosis and MSC chondrogenic hypertrophy in physioxia [93]. The reduction in apoptosis within chondrogenic pellets of MSCs may be related to the upregulation of autophagy genes that could also be a potential reason for the countering effect of physioxia on cytokine-inhibited chondrogenesis. Thus, further investigations are required into how the PI3K/Akt/FOXO pathway can contribute to alleviating cytokine-inhibited MSC chondrogenesis under physioxia, particularly with respect to autophagy genes and whether it can be manipulated to provide a fully protective effect. Another forkhead box transcription factor, *FOXA*, has also been implicated in the control of cartilage hypertrophy and subsequent endochondral ossification; therefore, further investigation into MSC chondrogenesis in physioxia could also consider its influence on the process [146].

## 7. Future Directions and Outlook

A STRING analysis was performed to investigate the possible interactions between the physioxia responsive pathway genes and chondrogenic genes (Figure 4 and Figure 5) [147]. The plots describe the relationships between proteins based on experimental and theoretical results compiled and evaluated from various databases and predicts the probability of interactions between them (Figure 4 and Figure 5). The analysis indicates a strong interaction or discrete clouds between the PI3K/AKT1/FOXO genes and the chondrogenic genes. In the first instance, PI3KCA is the central player with strong relationships between AKT and the different *FOXO* and *FOXA* genes evaluated. In terms of chondrogenic genes, *COL2A1* and *ACAN* are the central players. These discrete clouds have no direct interactions with a common gene between them (Figure 4 and Figure 5). Potentially, one common link between the two clouds is TGF-β receptor I, as the analysis shows a probable connection with PI3KCA. Therefore, one pathway for future investigation is the canonical and non-canonical TGF-β pathways and observing whether the PI3K/Akt pathway is stimulated by this process [148]. Interestingly, the HIF genes are separated from the chondrogenic genes, with the only connection between *HIF-1α* and *SOX9*, as described in this review [34]. The analysis also supports the experimental findings for the interaction between the PI3K/Akt/FOXO pathway and *HIF-1α* (Figure 5) [121]. However, it is noteworthy that the analysis does not show a direct interaction between the *HIFs* and PI3K/Akt genes with respect to chondrogenic gene expression, whilst the probability that there is a relationship between these distinct clouds is low. Thus, the STRING evaluations demonstrate that further investigations are required in understanding the underlying mechanisms controlling physioxia MSC chondrogenesis.

Since the phenotype of MSC-derived chondrocytes is not optimal, alternative cell types are being considered for use in the treatment of early OA focal defects. Recent investigations in cartilage tissue engineering have begun to evaluate a cell population within the superficial layer of articular cartilage, termed articular cartilage progenitor cells (ACPs) [33,149,150,151,152]. ACPs are isolated via fibronectin adherence and can be clonally expanded to high population doublings without a loss in chondrogenic potential, unlike MSC populations that progressively lose their chondrogenicity at similar doublings [149,151]. Furthermore, clonal ACPs have reduced collagen X staining and markers for osteogenesis compared to bone marrow-derived MSCs [33,150]. Under physioxia, ACP clonal populations demonstrated a significant downregulation in *COL10A1* and *MMP13* relative to hyperoxia and also exhibited no collagen X staining and protein expression in all ACP clones. In contrast, hyperoxic ACPs showed a clonal variability in collagen X staining, whereby there was either no or abundant collagen X staining, depending upon the clone [33]. Thus, ACPs are an alternative and potentially optimal cell source for the treatment of early OA focal defects, and physioxic preconditioning helps stabilise the chondrocyte phenotype.

In summary, culturing MSCs at physioxia increases proliferation and induces a greater anabolic response during chondrogenesis. Furthermore, physioxia supresses hypertrophy as judged by the downregulation in *COL10A1* and *MMP13* expression, although it does not completely inhibit the process. Whether physioxic preconditioning provides beneficial effects upon implantation in vivo is not yet established. The underlying mechanisms controlling the physioxic response remain to be elucidated and may offer new targets for therapies if novel pathways are identified.

## Figures and Tables

**Figure 1 ijms-20-00484-f001:**
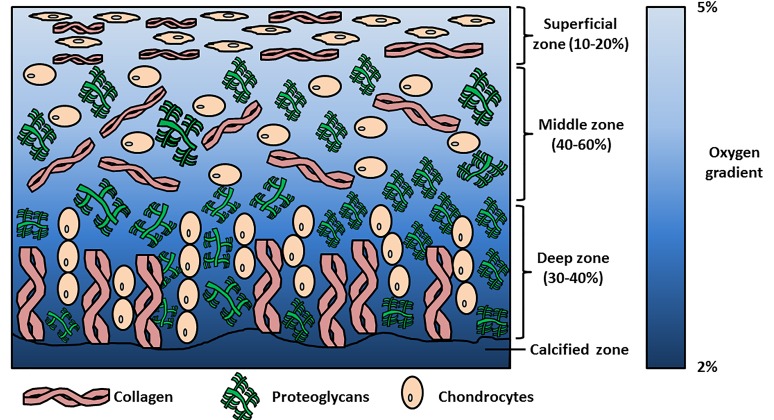
Schematic diagram describing the zones within articular cartilage and the changes in oxygen tension from the superficial zone to calcified zone.

**Figure 2 ijms-20-00484-f002:**
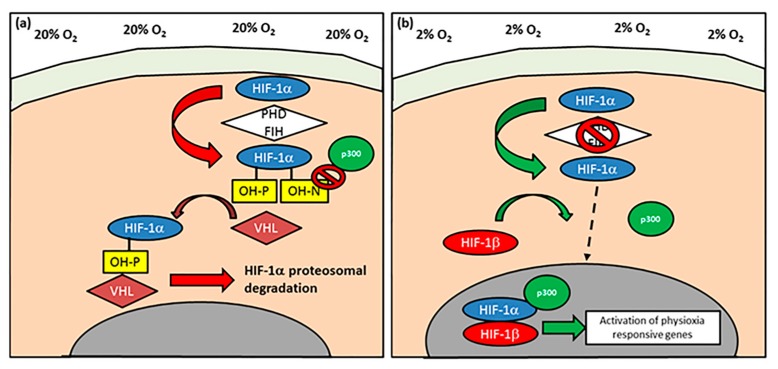
Schematic diagram describing the activation of *hypoxia-inducible factor-1 alpha* (*HIF-1α*). (**a**) Under normoxic/hyperoxic (20% oxygen) conditions, *HIF-1α* is hydroxylated by prolyl hydroxylases (PHDs) and factor inhibiting HIF (FIH) that enables *HIF-1α* to undergo proteasomal degradation by von-Hippel-Lindau (VHL) E3 ubiquitin ligase complex. In contrast, PHD and FIH activity is inhibited under (**b**) hypoxia/physioxia (2% oxygen), thus leading to nuclear translocation of *HIF-1α* that forms a complex with *HIF-1β* and co-factor p300, that results in upregulation of physioxia-responsive genes. Arrows describe the flow and stages involved in *HIF-1α* behavior under (**a**) normoxia/hyperoxia and (**b**) hypoxia/physioxia (dotted arrow depicts *HIF-1α* nuclear translocation).

**Figure 3 ijms-20-00484-f003:**
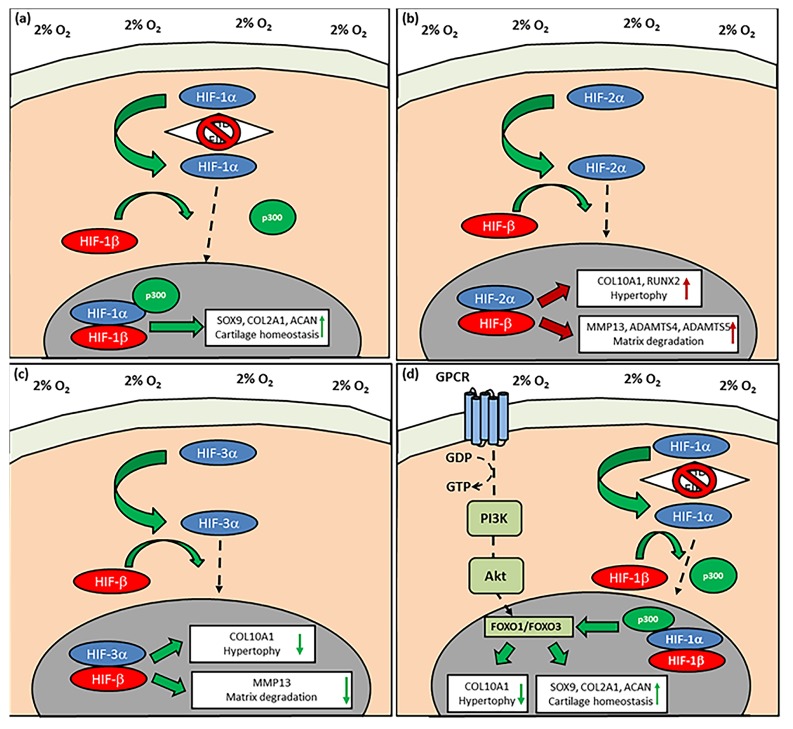
The pathways controlling the response of physioxia MSC chondrogenesis. (**a**) *HIF-1α* upon translocation into the nucleus and dimerization with *HIF-1β* leads to upregulation of chondrogenic matrix gene expression, whereas (**b**) *HIF-2α* nuclear translocation and dimerization results in an upregulation in cartilage hypertrophic and matrix degradation enzyme gene expression. The effect of *HIF-2α* can be countered by the upregulation of (**c**) *HIF-3α*, which upon activation, counters the upregulation in hypertrophic and matrix degradation gene expression. (**d**) The PI3K/Akt/FOXO pathway is activated under physioxia in response to *HIF-1α* translocation and helps to maintain chondrocyte phenotype via reduction in chondrocyte hypertrophy markers (green arrow direction symbolizes positive upregulation or downregulation in chondrogenic or hypertrophic genes; red arrow direction symbolizes negative upregulation in hypertrophic and matrix degradative enzyme gene expression; dotted arrow depicts *HIF-1α*, *HIF-2α* or *HIF-3α* nuclear translocation).

**Figure 4 ijms-20-00484-f004:**
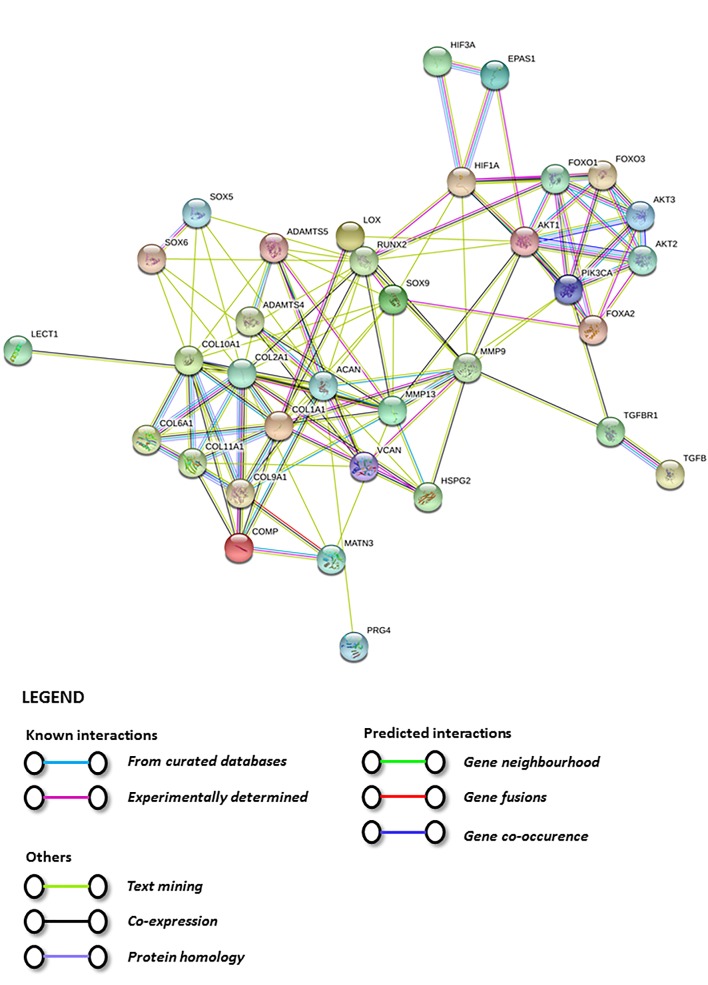
STRING database analysis based upon genes and proteins under the influence of physioxia and chondrogenesis, which describes the interactions based on the database analysis between proteins.

**Figure 5 ijms-20-00484-f005:**
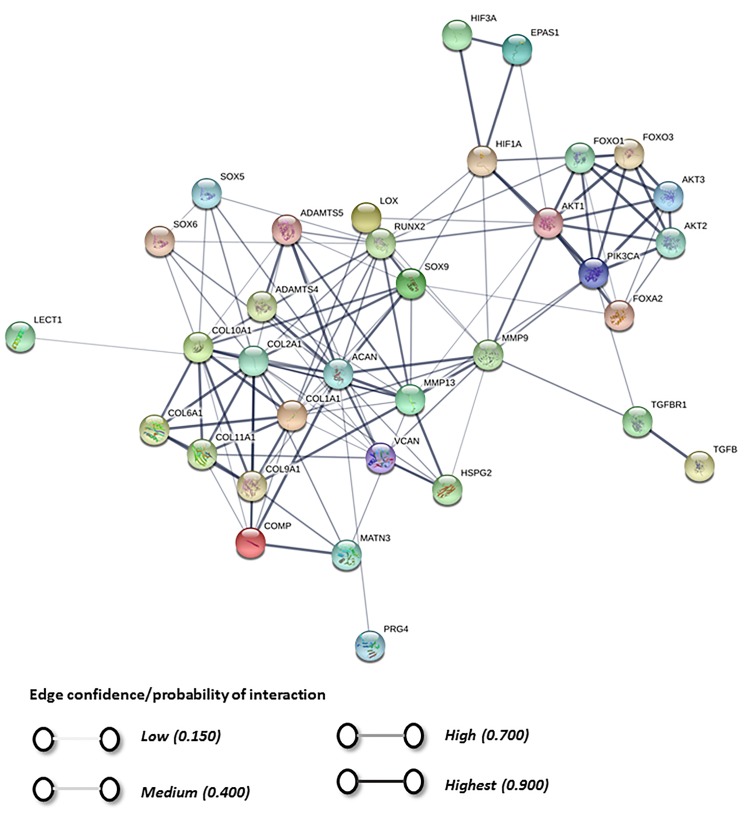
STRING database analysis based upon genes and proteins under the influence of physioxia and chondrogenesis, which describes the probability of interactions between proteins.

**Table 1 ijms-20-00484-t001:** Summary of the key findings from publications examining the effect of physioxia on mesenchymal stem cell (MSC) chondrogenesis. Findings are relative to hyperoxia.

Publication	Cell Source	Pellets or Scaffolds	Oxygen Tension	Results—Physioxia Response Relative to Hyperoxia
Robins et al., 2005 [61]	Mouse ST2 stromal cells/C3H10T1/2 cells	Pellet	1% O_2_	*Sex-determining region–box 9 (SOX9)* expression is upregulated and induces greater glycosaminoglycan (GAG) deposition. Increased *hypoxia-inducible factor-1 alpha* (*HIF-1α)* expression; no change in *hypoxia-inducible factor-1 beta (HIF-1β)* expression
Wang et al., 2005 [71]	Human adipose MSCs	Scaffolds (4 × 10^6^ cells/mL, alginate beads)	5% O_2_	Greater anaerobic respiration as measured by lactate production both under expansion and chondrogenesis. Increased GAG and collagen content
Betre et al., 2006 [65]	Human adipose MSCs	Scaffolds (2 × 10^6^ cells/scaffold; elastin-like polypeptide scaffold)	5% O_2_	Upregulation in *SOX9* and downregulation in *collagen X alpha 1* (*COL10A1*)—reduction in *collagen II alpha I* (*COL2A1*) and *aggrecan* (*ACAN*). No difference in GAG or collagen content between conditions
Malladi et al., 2006 [66]	Murine inguinal fat pad MSCs	Pellets	2% O_2_	Reduced GAG/DNA and collagen content but larger diameter pellets
Khan et al., 2007 [72]	Infrapatellar fat pad MSCs	Pellet	5% O_2_	*HIF1α*, *hypoxia inducible factor-2 alpha* (*HIF2α*), *SOX5*, *SOX6*, *SOX9*, *ACAN*, *COL2A1* and *COL10A1* increased expression. *HIF2α* had greater expression than *HIF1α*. Increased pellet wet weight, GAG content and collagen II staining
Malladi et al., 2007 [73]	Murine adipose MSCs (HIF-1α deleted mice)	Pellets	2% O_2_	*HIF-1α* expressed by adipose MSCs. *HIF-1α* deleted MSCs significantly decreased *SOX9*, *ACAN* and *COL2A1* expression. *HIF-1α* deleted micromasses had reduced GAG and collagen II deposition
Felka et al., 2009 [74]	Human bone marrow MSCs	Pellet	2% O_2_; 2 ng/mL IL-1β	No difference in gene transcript levels. Larger pellets with more matrix production. Physioxia increase chondrogenic gene (*SOX9*, *COL2A1*, *ACAN*), pellet size and matrix deposition with reduced expression in matrix metalloproteinase (*MMP1* and *MMP13*) in IL-1β inhibited chondrogenesis
Pilgaard et al., 2009 [63]	Adipose derived MSCs	Pellets	15%, 10%, 5%, 1% O_2_	*SOX9*, *collagen I alpha I* (*COL1A1*), *COL2A1* and *ACAN* upregulated at 15% oxygen and donwregulated under physioxia. Reduction in *COL10A1* expression. Increased matrix staining and GAG synthesis at 15% oxygen—reduced at lower oxygen tension. Increased matrix synthesis in central regions of ambient cultures due to oxygen gradients—central regions increase chondrogenesis in 15% oxygen culture
Baumgartner et al., 2010 [75]	Human bone marrow MSCs	Scaffold (20 × 10^6^ cells/ml fibrin hydrogel)	3% O_2_	Greater and earlier expression of *COL2A1*. Increased alcian blue matrix staining
Buckley et al., 2010 [76]	Porcine infrapatellar fat pad MSCs	Scaffold (15 × 10^6^ cells/mL in 2% agarose)	2% O_2_	Greater GAG and collagen II content with increased staining in core region. Superior mechanical properties
Khan et al., 2010 [77]	Human bone marrow MSCs	Pellet	5% O_2_	Upregulated *SOX6*, *COL2A1*, *ACAN*, *HIF1α* and *HIF2α* expression. Enhancement in pellet wet weight and GAG content
Merceron et al., 2010 [64]	Human adipose MSCs	Pellets	5% O_2_	*COL2A1* expression enhanced and no difference in *ACAN* expression. No difference in matrix deposition
Meyer et al., 2010 [78]	Porcine bone marrow MSCs	Scaffold (15 × 10^6^ cells/mL in 2% agarose)	5% O_2_	Greater GAG and collagen II content with increased staining in central regions. Increase in dynamic and equilibrium modulus. No synergistic effect with dynamic loading
Li and Pei, 2011 [55]	Human synovial fetal fibroblasts	Pellets	5% O_2_	*SOX9*, *ACAN* and *COL2A1* expression were upregulated. Larger pellets with greater GAG and collagen II content
Stoyanov et al., 2011 [79]	Human bone marrow MSCs	Scaffolds (4 × 10^6^ cells/mL in 1.2% (*w*/*v*) alginate beads)	2% O_2_	Increase in *SOX9* and *COL10A1* expression. Greater GAG and collagen II content. In the presence of GDF-5, increased *ACAN* and *COL2A1* expression compared to TGF-β groups with reduced hypertrophy
Gawlitta et al., 2012 [80]	Human bone marrow MSCs	Pellets	5% O_2_	Reduced collagen X staining
Meretoja et al., 2013 [67]	Bovine bone marrow MSCs	Scaffolds (Poly (ε-caprolactone; 4.5 × 10^6^ cells/mL, monoculture or co-culture (30% articular chondrocytes: 70% MSCs))	5% O_2_	*COL2A1* upregulated—further enhanced in co-culture. No difference in GAG and collagen content in MSCs monocultures—increased *alkaline phosphastase* (ALP) and calcification under these conditions. MSC-Chondrocyte co-cultures reduced MSC hypertrophy—chondrocytes prevent this process
Portron et al., 2013 [81]	Rabbit and human adipose MSCs	Pellets; Scaffolds (2 × 10^6^ cells/mL (rabbit) or 5 × 10^5^ cells/mL (human) in Si-HPMC)	5% O_2_	Upregulation in *COL2A1* and *ACAN* in both cell and culture types. Increased collagen II and GAG deposition. In vivo implantation of physioxia preconditioned scaffolds had higher O’Driscoll scores
Leijten et al., 2014 [82]	Human bone marrow MSCs	Pellets	2.5% O_2_	*SOX9*, *COL2A1* and *ACAN* gene expression upregulated and *COL10A1* and *MMP13* gene expression downregulated. Increased GAG staining for physioxia chondrogenesis. Physioxic preconditioned chondrogenesis reduced bone-like formation upon in vivo implantation
Munir et al., 2014 [70]	Human adipose MSCs	Pellets, Scaffolds (8 × 106 cells/mL in collagen type I/II scaffold–Chondroglide ^TM^)	5% O_2_	*SOX9*, *COL1A1* and *COL2A1* expression upregulated with downregulated *COL10A1*. Increase in *COL2A1*/*COL1A1* and *COL2A1*/*COL10A1* ratios. Increased matrix staining at periphery and more core deposition in hyperoxic pellets—same in scaffolds
Zhu et al., 2014 [83]	Human bone marrow MSCs	Scaffold (20 × 10^6^ cells/mL; Hyaluronic acid hydrogel)	1% O_2_	Reduced hypertophic marker (*COL10A1*, *MMP13*, *ALP*) expression in low cross-linking hydrgoels. Increased GAG content. High cross-linking density and hyaluronic acid concentration increased expression of hypertrophy markers
Portron et al., 2015 [84]	Human adipose MSCs	Pellets	5% O_2_	*SOX9*, *ACAN* and *COL2A1* upregulation and downregulation of *COL10A1* and *MMP13*. No difference in matrix staining
Markway et al., 2016 [85]	Human bone marrow MSCs	Pellets	2% O_2_; 7 days ± TNF-α (1 ng/mL) at 2% or 20% O_2_	Reduction in TNF-α generated loss in GAG content. Reduced *MMP2*, *MMP9* and *MMP13*; *ADAMTS4/5* expression. TNF-α inhibited MSC chondrogenesis
Galeano-Garces et al., 2017 [86]	Human adipose MSCs	PCL scaffolds and pellets	2% O_2_	*HIF1A*, *SOX9*, *COL10A1* and *indian hedgehog* (*IHH*) were significantly upregulated and *COL1A1* downregulated in pellet cultures. *SOX9* and *ACAN* expression had increased PCL scaffolds, whilst *COL10A1* expression was higher in hyperoxic cultures
Legendre et al., 2017 [87]	Human bone marrow MSCs	Collagen I/III sponges; TGF-β and BMP2 chondrogenic induction	3% O_2_	Significant upregulation in *COL2A1* and an increase in *COL2A1/COL1A1* and *COL2A1/COL10A1* ratio
Gomez-Leduc et al., 2017 [69]	Human umbilical cord MSCs	Collagen I/III sponges; TGF-β and BMP2 chondrogenic induction	5% O_2_	Lower expression of chondrogenic genes (*SOX9*, *COL2A1* and *ACAN*). Downregulation in *COL10A1* and *MMP13* expression, and reduced collagen X protein expression. Change in oxygen tension from hyperoxia (Day 0–7) followed by physioxia (Day 7–21) helped to stabilise chondrogenic phenotype with reduction in hypertrophic gene expression (*COL10A1*, *MMP13*)
Rodenas-Rochina et al., 2017 [88]	Porcine bone marrow MSCs	Polycaprolactone (PCL) composite scaffolds and PCL-hyaluronic acid coated scaffolds	5%O_2_	Significant increase in GAG deposition—no difference in collagen content. Greater collagen II staining
Bae et al., 2018 [47]	Human synovium MSCs	Pellets	5% O_2_	Significant upregulation in *SOX9*, *COL2A1* and *ACAN*. Downregulation in *COL10A1.* Increased GAG deposition and collagen II protein expression with reduced collagen X expression
Desance et al., 2018 [68]	Equine umbilical cord MSCs	Collagen I/III sponges; TGF-β and BMP2 chondrogenic induction	3% O_2_	No difference in chondrogenic gene (*SOX9*, *COL2A1* and *ACAN*) or hypertrophy gene (*COL10A1*, *runt-related transcription factor-2* (*RUNX2*)) expression. Hypertrophic genes were expressed significantly lower than chondrogenic genes

**Table 2 ijms-20-00484-t002:** Summary of key findings from publications examining the effect of physioxia preconditioning on MSC chondrogenesis. Findings are relative to hyperoxia.

Publication	Cell Source	Pellets or Scaffolds	Oxygen Tension	Physioxia Chondrogenic Response Relative to Hyperoxia
Martin-Rendon et al., 2007 [89]	Bone marrow MSCs	Pellets	1.5% O_2_	Upregulated and stabilised *HIF-1α* expression. Increase in *SOX9* gene expression and pellet wet weight
Xu et al., 2007 [49]	Murine adipose MSCs	Pellets	2% O_2_	*COL2A1* upregulated; no difference in *SOX9* and *ACAN* gene expression. Downregulation in MMPs (*MMP2*, *MMP3*, *MMP13*) and osteogenic genes (*RUNX2*, *ALP*). Physioxia preconditioning increased proteoglycan deposition—no influence of reoxygenation
Krinner et al., 2009 [41]	Ovine bone marrow MSCs	Pellets	5% O_2_	Enhancement in GAG and collagen II content
Markway et al., 2010 [90]	Human bone marrow MSCs	Pellets	2% O_2_	ACAN, *COL2A1* and *COL10A1* upregulated. Increased GAG content and larger pellets
Ronziere et al., 2010 [91]	Human bone marrow MSCs and adipose MSCs (only preconditioned)	Pellets	2% O_2_	No difference in *COL2A1* and *ACAN* expression. Reduction in hypertophic markers (*COL10A1* and *MMP13*) in physioxia preconditioned MSCs
Muller et al., 2011 [62]	Human bone marrow MSCs	Pellets, Scaffolds (4 × 10^5^ cells in 10% (*w*/*v*) gelatin)	4% O_2_	Upregulation in *SOX9*, *COL2A1*, *ACAN* and *COL10A1* in pellets and scaffolds. Larger pellets and increased GAG content
Weijers et al., 2011 [46]	Human adipose MSCs	Pellets	1% O_2_	*SOX9* and *COL2A1* upregulated. Increase in GAG content
Adesida et al., 2012 [42]	Human bone marrow MSCs	Pellets	3% O_2_	Upregulation in *SOX9*, *COL2A1* and *ACAN*; downregulation in *COL10A1*. Enhanced GAG content and collagen II staining. Increase in transforming growth factor–beta receptor one and two (*TGFBR1* and *TGFBR2*) and *HIF-2α* expression
Duval et al., 2012 [92]	Human bone marrow MSCs	Scaffolds (5 × 10^6^ cells/mL in alginate beads)	5% O_2_	Increase in *SOX5*, *SOX6*, *SOX9*, *ACAN* and *COL2A1* gene expression and decrease in *COL10A1*, *RUNX2* and *ALP*. Greater GAG and collagen II content upon in vivo implantation. Application of HIF-1α dominant negative plasmid prevents anabolic response
Sheehy et al., 2012 [45]	Porcine bone marrow MSCs	Pellets; Scaffold (15 × 10^6^ cells/mL, 2% agaose)	5% O_2_	Increase in GAG and collagen in both pellets and scaffolds (develops a pericellular matrix). Reduction in *ALP*; suppression in hypertrophy
Lee et al., 2013 [93]	Human bone marrow MSCs	Pellets	2% O_2_	*SOX9*, *COL2A1* and *ACAN* expression upregulated and *COL10A1* and *RUNX2* downregulated. Matrix staining for GAG, collagen II and collagen X support findings. Reduced staining for apoptotic markers (Caspase -3 and -8). Akt and downstream targets, *FOXO1* and *FOXO3*, were phosphorylated—Inhibition of pathway, increased hypertrophy (*COL10A1* and *RUNX2*) and reduced response. No effect on hyperoxic chondrogenesis
O’HEireamhoin et al., 2013 [49]	Human infrapatellar fat pad MSCs	Pellets, scaffolds (20 × 10^6^ cells/mL in 2% agarose or fibrin)	5% O_2_	Increase in GAG and collagen II content in pellets. Only GAG deposition increased within scaffolds—reduced collagen X staining
Pattappa et al., 2013 [54]	Human bone marrow MSCs	Pellets	5% or 2% O_2_	No difference in GAG content
Ranera et al., 2013 [94]	Equine bone marrow MSCs	Pellets	5% O_2_	*SOX9*, *COL2A1*, *ACAN*, *cartilage oligomeric matrix protein* (*COMP*) gene expression upregulated. Physioxia preconditioned cells enhanced GAG content. *HIF-1α* expression increased with time
Boyette et al., 2014 [44]	Ovine bone marrow MSCs	Pellets	5% O_2_	Enhanced chondrogenesis in physioxia differentiated cells but inhibited differentiation for physioxia preconditioned cells
Kalpakci et al., 2014 [50]	Dermis isolated MSCs	Pellets	5% O_2_	Increased GAG and collagen content in physioxia preconditioned MSCs; collagen II content was greater under hyperoxia
Bornes et al., 2015 [43]	Ovine bone marrow MSCs	Scaffolds (1 × 10^7^ cells/cm^2^ on either collagen type I and esterified hyaluronic acid scaffolds)	3% O_2_	Upregulated *ACAN* and *COL2A1* gene expression and downregulation in *COL10A1*. Enhanced GAG and collagen II content in both scaffold types
Anderson et al., 2016 [33]	Human bone marrow MSCs	Pellets	2% O_2_	*COL2A1* and *ACAN* upregulated whilst *COL10A1* and *MMP13* downregulated depending upon chondrogenic capacity. Enhanced GAG production. MSCs with high chondrogenic capacity stained for collagen X inspite of physioxic culture
Henrionnet et al., 2016 [95]	Human bone marrow MSCs	Alginate beads	5% O_2_	Upregulated *SOX9*, *COL2A1*, *ACAN* and *COMP*, downregulated *RUNX2* and *ALP* expression. No change in *COL10A1* expression. Sequential hyperoxia then physioxia increased *COL2A1* and *ACAN* expression with reduction in *COL10A1* expression. No calcification
Hudson et al., 2016 [96]	Human MSCs	Collagen-alginate scaffold	5% O_2_	Greater GAG content and mechanical properties
Ohara et al., 2016 [53]	Human synovial derived MSCs	Pellets	5% O_2_	No difference in pellet wet weight or matrix staining
Yasui et al., 2016 [52]	Synovium MSCs	Scaffolds (Sheet–like construct, 4 × 10^5^ cells/cm^2^)	5% O_2_	Increase in *SOX9*, *ACAN* and *COL2A1* expression. Increased GAG and collagen II content
Bornes et al., 2018 [51]	Ovine bone marrow MSCs	HYAFF scaffolds	3% O_2_	No difference in *COL2A1* and *ACAN* gene expression but significant increase in GAG content after 14 days culture. Significant downregulation in *COL10A1* with concomitant increase in *COL2A1/COL10A1* ratio at day 4 and 14. No difference in cartilaginous tissue formation in preconditioned chondrogenic MSCs upon in vivo implantation
Lee et al., 2018 [97]	Human bone marrow MSCs	Pellets	1% O_2_	Upregulation in *SOX9*, *COL2A1* and *ACAN* expression. Increase in GAG deposition

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
