# Peer review of "The Importance of Physioxia in Mesenchymal Stem Cell Chondrogenesis and the Mechanisms Controlling Its Response"

_ijms, 2019, doi:10.3390/ijms20030484_

Reviewer 1 Report

The authors reviewed well the current knowledge of physioxia and its role in stem cells and their differentiation into chondrocytes. In addition the figures provided will help the reader to better understand the signaling pathways activated by physioxia and how these pathways
control chondrogenesis. Therefore I felt that this is an excellent review on an important topic that plays a major role in cartilage repair.

Author Response

Dear reviewer,

We thank you for your kind and positive response to our submitted manuscript. and look forward to further reviewers in the future.

Reviewer 2 Report

The article presents the role of low oxygen tension in cartilage regeneration. The paper is well written and organized.

Nevertheless, some minor points should be improved:

- Substitute “degeradation” with “degradation” (page 2, line 3)

- The following sentence is not clear, please revise: “The chondrogenic induction conditions used for MSCs can also cause redifferentiation of articular chondrocytes in either pellets or hydrogels” (page 2, line 37).

- the acronym “GAG” should be written in full (page 2, line 39).

-table I and table II: it is not clear how the cited articles have been found (databases, keywords) and if they have been “selected” (for example for pertinence) or not. Moreover, the typology of the article (in vitro, in vivo or clinical study) should be clearly indicated.

Author Response

Dear reviewer,

We thank the reviewer for his response to our manuscript. The following is the answers to his comments

- Substitute “degeradation” with “degradation” (page 2, line 3)

This has now been changed to degradation, as the per the review’s suggestion

- The following sentence is not clear, please revise: “The chondrogenic induction conditions used for MSCs can also cause redifferentiation of articular chondrocytes in either pellets or hydrogels” (page 2, line 37).

We acknowledge the reviewers comment regarding the sentence. The sentence has now been revised to the following,

"The use of growth factors (e.g. TGF-β), low oxygen tension and other stimuli for MSC chondrogenesis can also be used to cause the redifferentiation of articular chondrocytes in either pellets or hydrogels".

- the acronym “GAG” should be written in full (page 2, line 39).

 The full name, glycosaminoglycan, has been added to the text.

-table I and table II: it is not clear how the cited articles have been found (databases, keywords) and if they have been “selected” (for example for pertinence) or not. Moreover, the typology of the article (in vitro, in vivo or clinical study) should be clearly indicated.

We are in agreement with the reviewer’s comment. In response, additional sentences have been added to the introduction to provide information on how articles were chosen and cited in this review. The following sentence was added at the end of the introduction section,

“A Pubmed search was performed and the date of the last search was 31st October 2018. The keywords used for the gathering of relevant publications used the terms, “mesenchymal stem cells” AND “hypoxia” OR “chondrogenesis” OR “chondrocytes”. Publications since 2001 were evaluated for the purposes of this review.”

Additionally, in tables 1 and 2, a column has been added “In vitro/in vivo” and then for each article it is stated whether the investigation was either an in vitro and/or in vivo study